# Robust Low-Sidelobe Transmit Beamforming under Peak-to-Average-Power Ratio Constraint

**DOI:** 10.3390/s23094468

**Published:** 2023-05-04

**Authors:** Lingping Cai, Ruixue Chu, Zhoupeng Ding, Yang Zou, Hongtao Li

**Affiliations:** School of Electronic and Optical Engineering, Nanjing University of Science and Technology, Nanjing 210094, China; cailpii@163.com (L.C.); crxxx19@163.com (R.C.); dzp@njust.edu.cn (Z.D.); zou_yang@njust.edu.cn (Y.Z.)

**Keywords:** robust transmit beamforming, peak-to-average power ratio, sidelobe control

## Abstract

Transmit beamforming (TBF) provides the capability of focusing illuminating power in the desired directions while reducing the emitting power in undesired directions. It is significantly important in low-altitude and slow-speed small (LSS) radar, which usually suffers from heavy clutter and rapidly changing interference on the near-ground side. Due to nonideal factors such as an inaccurate target direction and array gain-phase error, the robustness of TBF is also necessary to consider in practical applications. In this paper, we provide a robust TBF method that enables sidelobe control in preset regions and possesses high transmit efficiency in virtue of the peak-to-average-power ratio (PAPR) constraint on transmit weights. To achieve robustness, a norm upper bound is introduced to limit the fluctuation of transmit weights, and the steering vector mismatch is also considered by using a spherical uncertainty set surrounding the nominal steering vector. As the proposed robust TBF is nonconvex because of the nonconvexity of both the objective function and constraints, we translate it into a series of convex subproblems via several kinds of convex relaxation schemes. In particular, based on the special structure of the objective function and constraints, the translation of the nonconvex problem into a tractable SOCP problem is realized by using the combination of the triangle inequality and Cauchy–Schwartz inequality. Numerical results demonstrate the improvement in the efficiency and robustness of the proposed TBF method in comparison with traditional TBF methods.

## 1. Introduction

With the persistently increasing complexity of the radar working environment, traditional analog beamforming technology has been unable to meet increasingly stringent requirements. The development and application of digital beamforming (DBF) technology has improved the performance of radar in harsh environments. Due to its high level of flexibility and interference suppression ability, DBF is now widely used in both radar transmitters and receivers. Many scientific achievements have been made in various fields, such as radar [1], wireless communication [2], medical imaging [3], radio astronomy [4], etc. With the gradual progress of digital technology, DBF-based transmitting technology has recently become one of the hot topics in array signal processing and is increasingly exploited in radar and communication systems.

In order to alleviate the adverse effects of uncertain system factors and harsh working environments on beamformers, improving the robustness of beamformers has become an important issue in recent years. In its infancy, the research on robustness mainly focused on receive beamforming. Several representative robust adaptive beamforming (RAB) methods have been widely studied, including sample matrix inversion (SMI) beamforming [5], diagonal loading (DL) [6], and eigenspace-based beamforming [7]. For example, several algorithms were developed based on the SMI algorithm [8] and are widely used because of their good real-time performance, fast convergence, and low computational complexity [9,10]. Although the SMI algorithm performs well for small-scale arrays since the number of observations usually greatly exceeds that of sensors and the RMB rule is always satisfied [11], the estimation error of the sample covariance matrix can significantly impair the performance of modular or massive-scale arrays due to finite training data, leading to a rise in the sidelobe level, the shift of the mainlobe, the degradation of the output signal-to-interference-plus-noise ratio (SINR), etc. In addition, some problems exist in the other two RAB methods mentioned above, such as the difficulty determining the loading factor in DL beamforming and the high requirement for the SINR in eigenspace-based beamforming.

On the other hand, the robustness of RAB is enhanced by introducing additional objective functions or constraints, such as the positive semi-definiteness (PSD) of the signal-plus-noise covariance matrix [12], the norm of steering vectors [13,14,15,16], and the stochastic distortionless response [17,18]. Several extensions of the standard Capon beamformer (SCB) have been provided that improve the robustness to uncertain steering vectors. To deal with the steering vector mismatch, worst-case performance optimization (WCPO) [19], which limits the desired signal steering vector to various uncertainty sets, was performed by modeling the uncertainty set as an ellipsoid or sphere with a specific look direction. However, it is worth noting that more sophisticated prior information has to be available when the uncertainty set is no longer simply spherical. The work in [20] introduced a norm constraint into SCB and thus provided norm-constrained Capon beamforming (NCCB) [21] to develop a generalized doubly constrained robust (GDCR) beamformer. Ref. [22] proved that the ripple control over the steering vector uncertainty set can be transformed into a norm constraint for the weight vector, and a doubly constrained robust Capon beamformer (DCRCB) [23] proved to be the preferred choice for applications requiring high SINRs by combining a constant-norm constraint with a spherical uncertainty set constraint. Inspired by these works, Jiang described the uncertainty set as a rhombus to represent the $l_{\infty }$-norm constraint and devise a computationally acceptable strategy for efficiently solving large-scale $l_{\infty }$-beamforming problems via ADMM [24]. However, these RAB methods rarely consider the influence of signal distortion and often consume large computational overhead.

In recent years, transmit beamforming (TBF) has been extensively studied in the design of radar and communication systems [25]. In order to maximize the transmit power and reduce the dynamic range of DACs, constant modular (CM) constraints are generally required, but this severely limits the degrees of freedom (DoF) of weights. With the relaxation of CM, the peak-to-average-power ratio (PAPR) now raises concerns. In the incipient stage, PAPR was mainly used in the waveform design of OFDM and MIMO radar signals to control the dynamic range of transmit waveforms [26,27,28]. In massive-scale arrays, constant or approximately consistent amplitude constraints are imposed on the analog part of hybrid TBF. Therefore, a PAPR constraint is introduced into TBF as a trade-off between the power efficiency and the mainlobe ripple, peak sidelobe level (PSL), or other indicators by relaxing the unimodular constraints [29]. However, this TBF method is not yet suitable for RAB. In [30], Li imposed a PAPR constraint on transmit weights so as to avoid excessive energy attenuation and reduce the ripple of transmit weights. It has strong robustness to the steering vector mismatch. However, this method did not consider the sidelobe level of the beampattern, which considerably affects the performance of some radar systems suffering from strong interferences from a given region, especially for low-altitude and slow-speed small (LSS) radars.

As is known, the detection performance and low interception of radar will be reduced if the transmit beampattern has a high sidelobe level. There now exist a large number of works on the control of the sidelobe level [31,32,33,34,35,36]. On the one hand, sidelobe control is realized by introducing the sidelobe-level constraint directly. In [31,32], the sidelobe levels were precisely controlled for a ULA and an arbitrary geometric array, respectively. Furthermore, an additional weight function was defined to effectively control the sidelobe level in [33]. However, this method is computationally expensive and only applicable to a limited scope. On the other hand, received data reconstruction and PSL minimization were used to control the sidelobe level and to improve the robustness simultaneously. In [34], matrix reconstruction was exploited to improve the robustness and accurately control the sidelobe level. In [35], the alternating direction method of multipliers (ADMM) was used to solve sidelobe control combination constraints, which can effectively control the mainlobe, sidelobe, and nulling. In [36], the ratio between the minimum mainlobe level and the maximum sidelobe level was maximized by using a customized ADMM-based algorithm.

Based on the above observations, robust TBF with sidelobe-level control (SLC) is lacking. However, it is imperative in certain application scenarios, such as LSS target detection. Generally, “LSS target” is a collective term for a variety of small craft and air drifters that fly at low or ultra-low altitudes and have slow speeds and other characteristics. Radar detection is now quite a popular means of LSS target detection compared with acoustic detection [37] and photoelectric detection [38] because of its advantages of all-time, all-weather operation and fast response time. However, it faces difficulties in the complex urban background where the target is located, such as fast-moving interference, strong ground clutter and noise, and a high-complexity environment, resulting in the low SNR of the radar echo signal and affecting the detection performance of the radar.

To address the above problem, this paper proposes a robust TBF with sidelobe-level control under a PAPR constraint. This robust TBF approach can effectively deal with rapid changes in ground-side interference by minimizing the PSL in the preset region. Meanwhile, we address the array steering vector mismatch and norm constraint on the weight vector, combined with the sidelobe-level constraint, to further improve the robustness of the array beamformer and then better meet the requirements of actual radar systems. To enhance the target detection performance, the PAPR constraint on transmit weights is introduced to maximize the transmit power efficiency. To resolve the proposed nonconvex problem, we translate it into a series of convex subproblems via convex relaxation. Numerical results show that the proposed TBF method can effectively reduce the sidelobe level on the ground side and satisfy the PAPR constraint, which markedly improves the transmit power efficiency. The main contributions of this paper are summarized as follows:Based on the detection requirements of LSS radar, we aim to minimize the PSL of the transmit beampattern in a preset region while maintaining the mainlobe gain at the same time to preserve the target detection performance. Specifically, to sufficiently suppress the high-power clutter and rapidly changing interference from the ground, we require the sidelobe levels on the ground side to be sufficiently low but place no limitations on the sidelobe levels on the other side.It is pointed out in [39] that TBF with better performance can be obtained by properly relaxing the CM constraint. Therefore, a PAPR constraint on transmit weights is introduced to ensure that the weight amplitude fluctuates within a certain range to improve the efficiency of transmit power.The steering vector mismatch is considered to further strengthen the sidelobe-level constraint, and a norm upper bound of transmit weights is set to limit the fluctuation degree of the weight vector to a certain extent, thus improving the robustness of TBF.Under the PAPR and norm constraints on transmit weights, we translate the proposed TBF problem into an SOCP problem through a series of convex relaxation processing and then solve it effectively. Numerical results verify that the proposed method can offer a lower PAPR than existing deterministic and adaptive TBF methods.

The rest of this paper is organized as follows. In Section 2, linear-array-based array radar is taken as an example to develop the signal model and problem formulation of the proposed robust TBF with PAPR and norm constraints. Section 3 solves the proposed TBF problem by translating it into an SOCP problem with a series of convex relaxations. Numerical results and a performance analysis are provided in Section 4. Finally, Section 5 summarizes the paper.

Notations: Throughout this paper, boldfaced upper-case and lower-case characters represent matrices and vectors, respectively. The superscripts ${\left(\cdot \right)}^{*},{\left(\cdot \right)}^{T}$, and ${\left(\cdot \right)}^{H}$ denote conjugate, transpose, and Hermitian transpose operators. The phase and modulus operators are written as ∠(·) and |·|. The operator E{·} denotes the mathematical expectation, and ${∥{}_{•}∥}_{2}$ stands for the Euclidean norm of a vector.

## 2. Signal Model and Problem Formulation

Consider a narrowband signal transmitted by a linear array with *N* elements located at known locations $d_{n}\in R,n=1,2,\ldots ,N$, representing the distance from the *n*-th element to the reference element, as shown in Figure 1. The transmit beampattern can be expressed as
(1)$$\begin{array}{cc} \; & y\left(t\right)=h_{1}\omega_{1}^{*}s\left(t\right)+h_{2}\omega_{2}^{*}s\left(t\right)e^{j2\pi d_{2}\sin\theta /\lambda }+\cdots \\ \; & +h_{N}\omega_{N}^{*}s\left(t\right)e^{j2\pi d_{N}\sin\theta /\lambda }\\ \; & =\omega^{H}a\left(\theta \right)s\left(t\right) \end{array}$$
where $\omega ={\left[\omega_{1},\omega_{2},\ldots ,\omega_{N}\right]}^{T}$ is the transmit weight vector, and a(θ) is the array steering vector in the signal of interest (SOI) direction θ, which is defined as
(2)$$a\left(\theta \right)={\left[h_{1}\left(\theta \right),h_{2}\left(\theta \right)e^{j2\pi d_{2}\sin\theta /\lambda },\cdots ,h_{N}\left(\theta \right)e^{j2\pi {\dot{d}}_{N}\sin\theta /\lambda }\right]}^{T}$$
where λ is the wavelength, $h_{n}\left(\theta \right),n=1,2,\ldots ,N$ is the pattern radiated by the *n*-th element, and $h_{n}\left(\theta \right)=1$ when the element is isotropic. Then, the transmit beampattern in (1) can be expressed as
(3)$$F\left(\theta \right)=\omega^{H}a\left(\theta \right)$$

In the practical situation of LSS target detection, there are large amounts of clutter and interference existing on the ground, which increases the difficulty of LSS target detection. Therefore, we need to design an array weight vector to markedly suppress the sidelobe levels of the transmit beampattern on the near-ground side so that the reflected ground clutter can be significantly reduced. Based on this consideration, the method in this paper aims to minimize the PSL on the ground side while preserving the array response in the SOI direction. Then, the corresponding TBF problem can be formulated as the following min-max optimization problem:(4)$$\begin{array}{cc} \min_{\omega }\max_{\theta_{q}\in \Theta_{G}} & \left|\omega^{H}a\left(\theta_{q}\right)\right|\\ \;s.t.\; & \left|\omega^{H}a\left(\theta_{0}\right)\right|=1 \end{array}$$
where $\Theta_{G}$ indicates the direction set corresponding to the ground-side clutter region, $a\left(\theta_{q}\right)$ is the array steering vector in the direction $\theta_{q}$, and $\theta_{q}\in \Theta_{G}$.

In practice, the element spacing of the linear array may not be exactly identical because of the measurement error. Therefore, the actual gain and phase of the signals of each element are also in error with the ideal situation, which leads to the mismatch of the steering vector. The mismatched array steering vector inevitably affects the performance of the beampattern and leads to the degradation of the spatial filtering performance of the array.

In fact, if the gain and phase errors resulting from inaccurate element spacing and nonideal RF chains are incorporated into the array element weight vector, the actual beam response is
(5)$$\hat{F}\left(\theta \right)={\hat{\omega }}^{H}\hat{a}\left(\theta \right)=\sum_{n=1}^{N}{\hat{g}}_{n}e^{j\left({\hat{\varphi }}_{n}-2\pi {\hat{p}}_{n}\sin\left(\theta \right)/\lambda \right)}$$
where $\hat{\omega }$ and $\hat{a}\left(\theta \right)$ are the actual array weighting vector and the actual array steering vector, ${\hat{g}}_{n}$ is the actual amplitude weighting factor, ${\hat{\varphi }}_{n}$ is the actual phase weighting factor, and ${\hat{p}}_{n}$ is the actual position of the *n*-th array element.

When the above error is small, the actual amplitude-squared expectation of the beam response is
(6)$$\begin{array}{cc} E\left\{|\hat{F}{\left(\theta \right)|}^{2}\right\} & ={\left|F\left(\theta \right)\right|}^{2}e^{-\left(\sigma_{\varphi }^{2}+\sigma_{\lambda }^{2}\right)}+\left(\sigma_{\varphi }^{2}+\sigma_{\lambda }^{2}+\sigma_{g}^{2}\right)\sum_{n=1}^{N}g_{n}^{2}\\ \; & ={\left|F\left(\theta \right)\right|}^{2}e^{-\left(\sigma_{\varphi }^{2}+\sigma_{\lambda }^{2}\right)}+\left(\sigma_{\varphi }^{2}+\sigma_{\lambda }^{2}+\sigma_{g}^{2}\right){∥\omega ∥}^{2} \end{array}$$
where $\hat{F}\left(\theta \right)$ is the actual beampattern, and $\sigma_{g}$ and $\sigma_{\varphi }$ are, respectively, the variance of the amplitude and the phase error of the weights. $\sigma_{\lambda }=2\pi \sigma_{P}/\lambda$, where $\sigma_{P}$ is the variance of the antenna position error.

Equation (Equation 6) consists of two parts. The first part is the ideal beampattern multiplied by an attenuation coefficient, which will not affect the gain of beam formation. The second part is the product of the sum of the errors and the norm of the array weighting vector, which mainly affects the sidelobe level of the beampattern.

The sensitivity function is defined as
(7)$$T_{\mathrm{se}}=\sum_{n=1}^{N}g_{n}^{2}={∥\omega ∥}^{2}$$

Therefore, for a certain error, the smaller the ${∥\omega ∥}^{2}$, the higher the robustness of beamforming, and ${∥\omega ∥}^{2}$ can be used as an important indicator for the robustness of beamforming. In order to design a robust beamformer, ${∥\omega ∥}^{2}$ should be small. Therefore, we set the following robust constraint:(8)$${∥\omega ∥}^{2}\leq b$$
where *b* is the upper bound of the prescribed norm, and different values can be specified according to the actual requirements.

In addition to improving robustness, maximizing the transmit power is important to improve the reliability of the sensing system. It is well known that receive and transmit beam designs are reciprocal because of the transmit–receive reciprocity of the antenna. In practice, however, the transmit beam design becomes more difficult due to the energy and PAPR constraints. Ideally, to obtain a stronger signal-to-noise ratio (SNR) with better performance, a radar transmitter for detecting LSS targets should transmit all the energy to the area where the target is located. For TBF, to maximize the radar transmit power, the transmit signal needs to meet the CM constraint or have a low PAPR.

To obtain a better performance of the transmit beam and to avoid excessive energy fading, we introduce a PAPR constraint on the transmit weights. The low-PAPR constraint ensures that the amplitude factor fluctuates within a certain range while improving the utilization of the transmit signal power. According to the definition of the PAPR, the constraint on transmit weights can be expressed as
(9)$$r_{\mathrm{PAPR}}=\frac{\max_{i=1,2,\cdots ,N}{\left|\omega_{i}\right|}^{2}}{{∥\omega ∥}^{2}/N}=N\frac{\max_{i=1,2,\cdots ,N}{\left|\omega_{i}\right|}^{2}}{{∥\omega ∥}^{2}}$$

Considering the low-PAPR constraint, the preservation of the mainlobe response, and the robustness norm constraint, problem (Equation 4) can be further expressed as
(10)$$\begin{array}{cc} \min_{\omega }\max_{\theta_{q}\in \Theta_{G}} & \left|\omega^{H}a\left(\theta_{q}\right)\right|\\ \;s.t.\; & \left|\omega^{H}a\left(\theta_{0}\right)\right|=1\\ \; & r_{\mathrm{PAPR}}\leq \zeta \\ \; & \omega^{H}\omega \leq b \end{array}$$
where ζ is the preset upper bound of PAPR tolerance.

## 3. The Proposed Algorithm

In this section, a robust low-sidelobe TBF method under a low-PAPR constraint and a robust constraint is proposed.

Note that the PAPR constraint $r_{\mathrm{PAPR}}\leq \zeta$ in (Equation 10) is difficult to handle since it is a fractional constraint related to ω. Therefore, we first convert this constraint into a substitutive expression that is tractable.

Since $a{\left(\theta_{0}\right)}^{H}\omega =1$, we have the following inequality:(11)$$1={\left|a^{H}\left(\theta_{0}\right)\omega \right|}^{2}\leq {∥a\left(\theta_{0}\right)∥}^{2}{∥\omega ∥}^{2}=N{∥\omega ∥}^{2}$$

Therefore, we can obtain
(12)$${∥\omega ∥}^{2}\geq 1/N$$

By using (Equation 12), the PAPR constraint can be rewritten as
(13)$$r_{\mathrm{PAPR}}=N\frac{\max_{i=1,2,\cdots ,N}{\left|\omega_{i}\right|}^{2}}{{∥\omega ∥}^{2}}\leq N\frac{\max_{i=1,2,\cdots ,N}{\left|\omega_{i}\right|}^{2}}{1/N}=N^{2}\max_{i=1,2,\cdots ,N}{\left|\omega_{i}\right|}^{2}\leq \zeta$$

In other words,
(14)$$N^{2}\max_{i=1,2,\cdots ,N}{\left|\omega_{i}\right|}^{2}\leq \zeta$$
which is equivalent to
(15)$$\max_{i=1,2,\cdots ,N}{\left|\omega_{i}\right|}^{2}\leq \sqrt{\zeta }/N$$

Obviously, the PAPR constraint is substituted by many convex constraints. Then, problem (Equation 10) is converted into
(16)$$\begin{array}{cc} \min_{\omega }\max_{\theta_{q}\in \Theta_{G}} & \left|\omega^{H}a\left(\theta_{q}\right)\right|\\ \;s.t.\; & \left|\omega^{H}a\left(\theta_{0}\right)\right|=1\\ \; & \left|\omega_{i}\right|\leq \sqrt{\zeta }/N,\;\forall i\\ \; & \omega^{H}\omega \leq b \end{array}$$

In practical beamforming applications, the assumed signal steering vector often suffers from a certain error; i.e., there is a certain mismatch between the assumed signal steering vector and its actual value. In order to improve the robustness of beamforming against an arbitrary steering vector mismatch, the actual steering vector is defined as
(17)$$\hat{a}=a+a_{e}$$
where $a_{e}$ is the steering vector error. The uncertainty region of $\hat{a}$ can be modeled as a ball set ε, namely,
(18)$$\varepsilon =\left\{a_{e}|∥a_{e}∥\leq \varepsilon_{e}\right\}$$
where $\varepsilon_{e}$ is the radius of the ball. Then, problem (Equation 16) is translated into
(19)$$\begin{array}{cc} \min_{\omega }\max_{\theta_{q}\in \Theta_{G},\hat{a}\left(\theta \right)\in \varepsilon } & \left|\omega^{H}\hat{a}\left(\theta_{q}\right)\right|\\ \;s.t.\; & \left|\omega^{H}a\left(\theta_{0}\right)\right|=1\\ \; & \left|\omega_{i}\right|\leq \sqrt{\zeta }/N,\;\forall i\\ \; & \omega^{H}\omega \leq b \end{array}$$

Furthermore, problem (Equation 19) can be equivalent to
(20)$$\begin{array}{cc} \min_{\omega }\max_{\theta_{q}\in \Theta_{G}}\max_{\hat{a}\left(\theta \right)\in \varepsilon } & \left|\omega^{H}\hat{a}\left(\theta_{q}\right)\right|\\ \;s.t.\; & \left|\omega^{H}a\left(\theta_{0}\right)\right|=1\\ \; & \left|\omega_{i}\right|\leq \sqrt{\zeta }/N,\;\forall i\\ \; & \omega^{H}\omega \leq b \end{array}$$

Note that (Equation 20) is a nonconvex constrained optimization problem due to the nonconvex objective function, and its objective function has an infinite dimension. In the following, we will translate (Equation 20) into a tractable convex problem by using the special structure of the objective function and constraints.

Using the triangle inequality and Cauchy–Schwartz inequality, a tight upper bound of $\left|\omega^{H}\hat{a}\left(\theta_{q}\right)\right|$ is determined as
(21)$$\left|\omega^{H}\hat{a}\left(\theta_{q}\right)\right|=\left|\omega^{H}\left(a\left(\theta \right)+a_{e}\left(\theta \right)\right)\right|\leq \left|\omega^{H}a\left(\theta \right)\right|+\left|\omega^{H}a_{e}\left(\theta \right)\right|$$

According to (Equation 21) and $∥a_{e}∥\leq \varepsilon_{e}$, we have
(22)$$\max_{∥a_{e}|\leq \varepsilon_{e}}\left|\omega^{H}\left(a\left(\theta \right)+a_{e}\left(\theta \right)\right)\right|\leq \left|\omega^{H}a\left(\theta \right)\right|+\varepsilon_{e}∥\omega ∥$$
where the equality holds only if $a_{e}\left(\theta \right)=\varepsilon_{e}e^{j\varphi }\omega /∥\omega ∥$ with $\varphi =\mathrm{angle}\left\{\omega^{H}a\left(\theta \right)\right\}$. The left-hand side of (Equation 21) has two terms, one for the ideal pattern function and the other for beam robustness.

According to inequality (Equation 22), we can reformulate problem (Equation 20) into
(23)$$\begin{array}{cc} \min_{\omega }\max_{\theta_{q}\in \Theta_{G}} & \left|\omega^{H}a\left(\theta_{q}\right)\right|+\varepsilon_{e}∥\omega ∥\\ \;s.t.\; & \left|\omega^{H}a\left(\theta_{0}\right)\right|=1\\ \; & \left|\omega_{i}\right|\leq \sqrt{\zeta }/N,\;\forall i\\ \; & \omega^{H}\omega \leq b \end{array}$$

With the combination of the triangle inequality and Cauchy–Schwartz inequality, problem (Equation 20) is finally transformed into the following convex optimization problem:(24)$$\begin{array}{cc} \min_{\omega ,t}\; & t+\varepsilon_{e}∥\omega ∥\\ \;s.t.\;\; & \left|\omega^{H}a\left(\theta_{0}\right)\right|=1\\ \; & \left|\omega_{i}\right|\leq \sqrt{\zeta }/N,\;\forall i\\ \; & \omega^{H}\omega \leq b\\ \; & \left|\omega^{H}a\left(\theta_{q}\right)\right|\leq t,\theta_{q}\in \Theta_{G} \end{array}$$
where *t* is an auxiliary variable.

Note that (Equation 24) is an SOCP problem, which is a subclass of the convex programming problem. We can therefore solve it by using off-the-shelf convex optimization toolboxes, such as CVX.

## 4. Numerical Results and Analyses

This section describes several representative experiments that were performed to evaluate the effectiveness of the proposed robust TBF method. The following experiments are based on a ULA, whose structure is shown in Figure 1. Taking the first array element as the reference element, suppose that the ULA has N=32 isotropic radiation elements, and there is no mutual coupling effect among the elements. To prevent the appearance of grating lobes, the element spacing is set at a half wavelength. It is assumed that the mainlobe direction of the beampattern is $\theta_{0}=0^{\circ }$, the scanning range is $\left[-90^{\circ },90^{\circ }\right]$, and the sampling interval is $0.1^{\circ }$. The sidelobe regions are $\left[-90^{\circ },-5^{\circ }\right]$ on the ground side and $\left[5^{\circ },90^{\circ }\right]$ on the air side. Training snapshots were set as 500 for adaptive TBF. The convex problems were solved by using the CVX toolbox. The basic parameter settings of the numerical experiments are shown in Table 1.

In the follow-up experiments, we first compared the transmit beampatterns generated by the proposed method and other typical beamforming methods, including SMI beamforming, static beamforming, NCCB beamforming, and WCPO beamforming, so as to evaluate the effectiveness of the proposed TBF method. Then, to quantize the influence of the PAPR constraint and norm constraint on TBF, the performance was studied under different parameter configurations, and simulation experiments were conducted with different parameter settings.

Two typical metrics are introduced to measure the performance of sidelobe control, namely, the PSL and the ISL. The PSL, which the proposed method aims to minimize, measures the ability of the radar system to distinguish neighboring targets. On the other hand, the ISL measures the impact of the whole sidelobe region in the desired target direction. The smaller the PSL and ISL, the lower the energy of the sidelobe returned by the strong clutter and undesired targets, and the higher the probability of detecting the weak desired target, thus improving the detection performance. The following formulas can be used to calculate the two metrics.
(25)$$\mathrm{PSL}=10\log\left(\frac{\max_{\theta \in \Theta_{G}}{\left|\omega^{H}a\left(\theta \right)\right|}^{2}}{{\left|\omega^{H}a\left(\theta_{0}\right)\right|}^{2}}\right)$$
and
(26)$$\;\mathrm{ISL}\;=10\log\left(\frac{\int_{\theta \in \Theta_{G}}{\left|\omega^{H}a\left(\theta \right)\right|}^{2}}{{\left|\omega^{H}a\left(\theta_{0}\right)\right|}^{2}}\right),\theta \in \Theta_{G}$$

### 4.1. Experiment 1: The Proposed TBF versus Several Typical Beamformers

In this experiment, we compared the transmit beampatterns provided by the proposed method and several typical counterparts. As is known, traditional TBF methods are often sensitive to inaccurate covariance matrix estimation. With few snapshots, the sampling covariance matrix can only provide a coarse covariance matrix estimate. In addition, the direction of arrival of interferences, which are close to the mainlobe, can hinder the formation of notches and reduce the gain of the mainlobe, thus causing the significant deterioration of the TBF performance.

In this experiment, we assumed that 80 interferences exist in a continuous region on the ground side. The specific parameters are summarized in Table 2. According to the RMB rule [11], fully adaptive processing requires the number of snapshots to be not less than twice the system’s degrees of freedom, i.e., 2N. Based on this principle, Figure 2 displays the transmit beampatterns generated by the proposed TBF method and four typical counterparts with different numbers of snapshots. The detailed PSL and ISL of transmit beampatterns of these TBF methods are shown in Table 3, where the data before and after ‘,’ correspond to 64 and 500 snapshots, respectively.

As shown in Figure 2b, when the number of snapshots is large enough, traditional adaptive beamforming has excellent performance. However, in practice, the sampling frequency is often limited, as the larger the sampling rate is, the higher the data rate is, which will impose a great burden on the storage and processing of the system. As shown in Figure 2a,c, when interferences change rapidly, such as in the vehicle-mounted environment, the performance of the adaptive methods will drastically decrease due to insufficient snapshots. The smaller the snapshots, the more the performance decreases. The results of this experiment are shown in Table 3.

According to Table 3, although deep notches against interferences can be formed, the PSLs and ISLs on both sides are much higher than those of the proposed method in most cases after a further considerable rise when the snapshots decrease to 64. In addition, the actual PAPRs of the three adaptive methods are quite large compared with that of the proposed method, which increases the power consumption. Static beamforming and the proposed method are not adaptive, which means they cannot be affected by snapshots, which is why the data of these two remain consistent. However, the former can neither effectively suppress the interference nor suppress the sidelobe level.

In addition to the snapshots, the DOAs of interferences also affect the TBF performance, such as the gain, notch depth, and so on. Considering that the interference region is $\left[-59.5^{\circ },-20^{\circ }\right]$, which is far away from the mainlobe, and $\left[-44.5^{\circ },-5^{\circ }\right]$, which is close to the mainlobe, the interference interval is $0.5^{\circ }$, the INRs are set as 40 dB, and other parameters are unchanged. Figure 3 shows the TBF beampatterns formed by all five methods when the interference region is close to the mainlobe. Note that the mainlobes of SMI and WCPO are offset by $3.1^{\circ }$, and the gains are −5.99dB. Compared with the beampatterns in Figure 2a,b, SMI and WCPO can still form deep notches against the interferences. However, the sidelobe levels on both sides significantly increase, especially when the snapshot is limited, leading to marked performance degradation.

As shown in Table 4, compared to SMI and WCPO, NCCB and static beamformers have a better mainlobe performance, and the sidelobe level facing the air is relatively low; however, NCCB’s sidelobe level is obviously much higher than that in Figure 2, and the notches are shallower to interference as well.

Compared with the four comparative methods, the proposed method can achieve a smaller PAPR and fully improve the transmission efficiency. It effectively suppresses the sidelobe level on the ground side and reduces the impact of interferences from the ground with high stability due to not being affected by the accuracy of the sampling covariance matrix. As the air area is relatively clean and the electromagnetic interference is relatively low, the requirement for interference suppression in the airspace is low. Even if the sidelobe level in the airspace is quite high, the transceiver performance of the mainlobe and the overall radar performance will not be significantly affected.

### 4.2. Experiment 2: PAPR Tolerance vs. Sidelobe Level

In this experiment, we revealed the relationship between the variation in the PAPR tolerance and the beampattern formed by the proposed method with different norm upper bounds of the weight vector. Here, as shown in Table 5, the PAPR tolerance range is set as [1.1,2], and the upper bound is varied to equal 0.0325,0.035,0.045, and 1.

Taking LSS target detection as an example, we need to avoid out-of-band radiation, which affects signals in adjacent frequency bands, and in-band distortion, which causes the rotation, attenuation, and displacement of signals. We also want the ability to handle large-scale and fast-moving interference on the ground. That is, we expect to obtain a low PAPR, a low SLL, and high robustness at the same time. However, these goals are mutually contradictory. Therefore, seeking for a compromise proposal to select the proper *b* is vital to the ultimate performance of the proposed method. Figure 4 shows beampatterns with different constant upper limits of the norm and varying PAPR tolerances. We further present Figure 5 in order to display the more explicit relationship between the PAPR and SLL.

As shown in Figure 5, when the upper limit value is infinitely close to the lower bound 1/N, the sidelobe level is high and the inhibition effect is relatively weak. In addition, with the relaxation of the PAPR constraint, the improvement effect on the sidelobe level is not very substantial. As the value of *b* gradually increases, the sidelobe inhibition effect gradually improves, and the PAPR constraint has far more influence on this improvement. This situation continues until b≈0.045. After that, the increase in *b* will no longer reduce the sidelobe level, and the influence degree of PAPR variation will no longer increase. According to the beampattern, when the sidelobe facing the ground side decreases, the counterpart on the air side will rise to a certain extent. However, the maximum sidelobe level is at least 10dB lower than the mainlobe level, so it will not significantly affect the transceiver performance of the mainlobe or the overall performance of radar.

## 5. Conclusions

This paper presents a robust TBF method with sidelobe-level control for phased-array radar under PAPR and norm constraints. To suppress the strong clutter and rapidly changing interference on the near-ground side for LSS radar, we developed a robust TBF problem to minimize the PSL in the preset region, where the PAPR constraint and the norm constraint are introduced to enhance the radar transmit efficacy and to improve the robustness, respectively. To further improve the robustness, we merged the steering vector mismatch into the proposed problem and translated the corresponding nonconvex problem into an SOCP problem by using a series of convex relaxations. Numerical results proved that the proposed method can effectively suppress the interference on the ground side due to the very low sidelobe level on the near-ground side and is robust to the steering vector mismatch, and the PAPR value has an effect on the sidelobe level. As the PAPR continues to decline, the sidelobe level is further suppressed; however, once the tolerance is greater than 1.3, its effect on the sidelobe level is almost negligible.

It is worth noting that the results obtained based on a fundamental model without considering mutual coupling are somewhat different from those in the actual situation. We may further consider more physical factors and apply the proposed digital TBF method to other situations. In cutting-edge applications such as automatic driving and UAV-mounted radar, small-scale digital arrays cannot precisely distinguish intensive targets and meet the real-time requirement due to their limited performance in terms of the detection range, angular resolution, and SNR. A large-scale array is then adopted to enhance the radar performance. To reduce the cost and hardware complexity, the employed large-scale array is usually not a full-digital array but has special structures, such as an analog phased array or a hybrid analog-and-digital array. Therefore, the corresponding transmit beamforming is equivalent to a sophisticated nonconvex constrained optimization problem because of the nonconvex constraints generated by the array structure and the beamforming requirement. Furthermore, to improve the array efficiency, antenna selection is now widely used to segment the whole array into several sparse sub-arrays for multi-functional applications, i.e., the coexistence of communication and radar systems. Along with the antenna selection, transmit beamforming should be translated into a sparsity-regularized nonconvex constrained optimization problem. In all of these cases, how to efficiently solve transmit beamforming problems is a key question. To sum up, our future work will explore the further development of the proposed method so that it can be applied to more complicated scenarios.

## Figures and Tables

**Figure 1 sensors-23-04468-f001:**
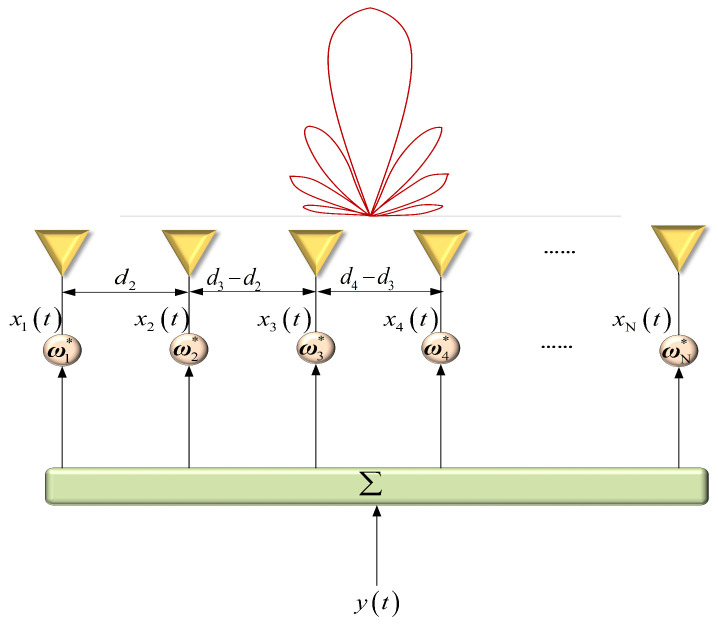
Transmit array layout of a linear array.

**Figure 2 sensors-23-04468-f002:**
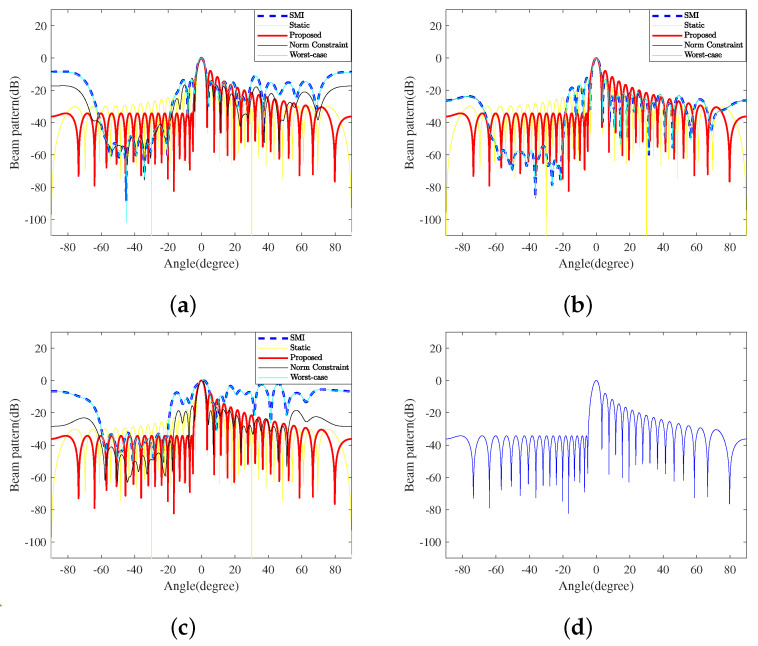
The beampatterns of the proposed TBF method and four typical counterparts with different numbers of snapshots *k*. (**a**) k=64; (**b**) k=500; (**c**) k=32; (**d**) the proposed beampattern in (**c**).

**Figure 3 sensors-23-04468-f003:**
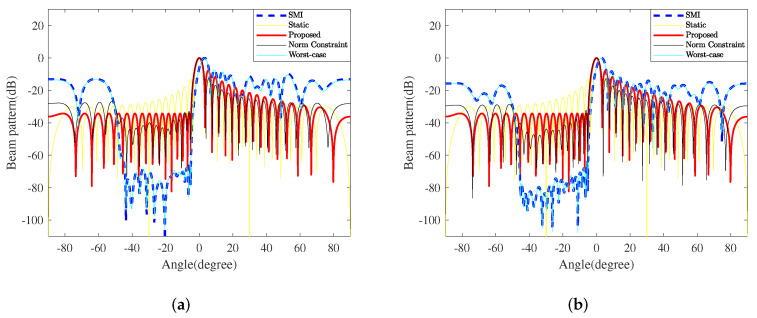
The beampatterns of the proposed TBF method and four typical counterparts when the interference region is close to the mainlobe. (**a**) k=64; (**b**) k=500.

**Figure 4 sensors-23-04468-f004:**
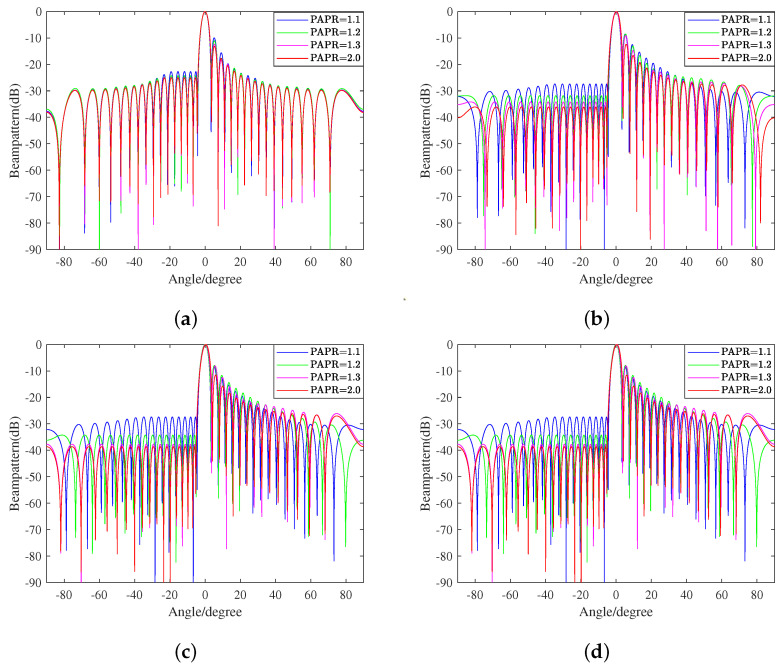
Beampatterns with different constant upper limits of the norm. (**a**) b=0.0325; (**b**) b=0.035; (**c**) b=0.045; (**d**) b=1.

**Figure 5 sensors-23-04468-f005:**
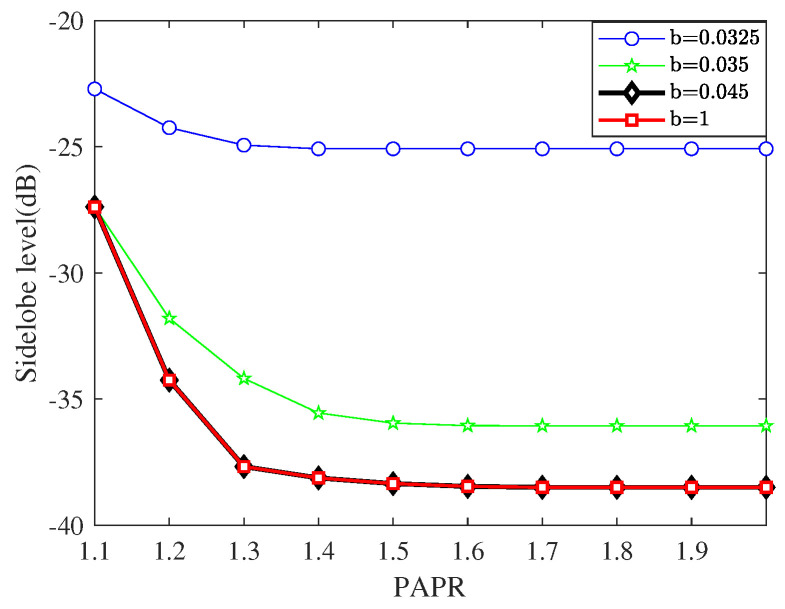
The sidelobe level with constant upper limits of the norm and different PAPR tolerances.

**Table 1 sensors-23-04468-t001:** General parameter settings.

Array Model	ULA	Element Spacing	λ/2
Scanning Range	$\left[-90^{\circ },90^{\circ }\right]$	Expected Signal	$\theta_{0}=0^{\circ }$
Sampling Interval	$0.1^{\circ }$	Ground Side	$\left[-90^{\circ },-5^{\circ }\right]$
Snapshots	500	Air Side	$\left[5^{\circ },90^{\circ }\right]$

**Table 2 sensors-23-04468-t002:** Parameter settings of comparison experiment.

PAPR	1.2	Norm Upper Bound	0.8
Interference direction	$-59.5^{\circ }:0.5^{\circ }:-20^{\circ }$	Snapshots	32, 64, 500
Interference number	80	INRs	10dB

**Table 3 sensors-23-04468-t003:** Detailed PSL and ISL of five TBF methods with different snapshots.

Method	PAPR	PSL (dB)		ISL (dB)		Norm
		Ground	Air	Ground	Air	
Static	1	−13.24	−13.24	2.02	2.02	5.66
SMI	3.43,1.88	−12.84,−12.90	−5.929,−12.89	6.83,3.02	14.18,4.09	0.22,0.19
Norm constraint	1.73,1.58	−13.16,−13.62	−11.50,−12.84	2.78,2.31	5.48,3.72	0.19,0.19
Worst case	3.42,1.87	−12.81,−12.92	−5.99,−12.89	6.79,3.01	14.06,4.07	0.22,0.19
Proposed method	1.03	−34.25	−7.89	−7.67	7.83	0.18

**Table 4 sensors-23-04468-t004:** The detailed information of different TBF methods.

Method	PAPR	PSL (dB)		ISL (dB)		Norm
		Ground	Air	Ground	Air	
Static	1	−13.24	−13.24	2.02	2.02	5.66
SMI	2.27,2.62	−13.05,−13.80	−3.28,−4.27	10.86,8.33	15.49,12.74	0.49,0.43
Norm constraint	1.53,1.53	−26.76,−28.59	−12.60,−12.86	−3.26,−4.70	3.89,3.37	0.19,0.19
Worst case	2.23,2.54	−13.30,−14.02	−3.41,−4.32	10.52,8.11	15.09,12.60	0.50,0.44
Proposed method	1.03	−34.25	−7.89	−7.67	7.83	0.18

**Table 5 sensors-23-04468-t005:** Settings of experiment 2.

PAPR	[1.1, 2]
Upper limit of norm	0.0325,0.035,0.045,1
Snapshots	4000

## Data Availability

Not applicable.

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
