# Peer review of "Robust Low-Sidelobe Transmit Beamforming under Peak-to-Average-Power Ratio Constraint"

_sensors, 2023, doi:10.3390/s23094468_

Round 1

Reviewer 1 Report

To get better transmit efficiency and sidelobe suppression performance, this document addresses the problem of robust transmit beamforming (TBF) under peak-to-average power ratio (PAPR) and sidelobe level constraints. For small, low-altitude, low-speed radar applications, minimizing the sidelobe peak level provides a robust TBF method. Particularly, the authors first transform the PAPR constraints into a tractable form and handle robust transmission by accounting for control vector mismatches in both the objective function and the weight-norm constraints. With a series of convex relaxations the process transforms the proposed non-convex constrained optimization into an easily solvable quadratic convex kernel programming problem. Numerical results demonstrate the efficiency and robustness of the proposed method compared with typical deterministic and adaptive TBF1 methods. There is moderate overlap, the authors are required to bring it down. There are some typos e.g. 'contraint', 'contrained' in the abstract, the authors are to correct them. Is a prototype made and experimentally validated? What is the physical basis of introducing two matrices(PSL and ISL)?

minor changes required

Reviewer 2 Report

The paper presents a TBF method with sidelobe control to increase transmission performance and robustness. I must admit that my experience in this field is limited to military radars from a part of my professional career where I worked in a NATO consortium, and although I am familiar with the subject matter, my assessment of the paper has to be considered overall as I cannot offer any criticism beyond obvious errors on the deeper mathematical content of the content. 

The article is reasonably brief and well presented, but there are methodological aspects that I believe need to be improved. The introduction is too direct, and not even a summary has been included to allow the reader to focus on the problem to be analyzed; I think it should be expanded and written in a progressive way. In the same way that the figures compare different methods, SMI, STATIC, etc., the introduction, some 90 lines, does not explain the characteristics of any of them. I think it is essential to introduce this comparison with its corresponding technical description.

The references seem to me to be poor, a good part of them, which I assume correspond to classic concepts, are 20 or more years old. It is essential to introduce recent references where studies similar to the one being studied in the article are presented. 

The document tackles simulation head-on and, although the results seem to justify a certain improvement, all that can be seen of the simulation, beyond the equations, is the figure of a Uniform Linear Array (ULA, not defined in a glossary) whose structure is conventional, despite the existence of publications with improvements with respect to the one presented. Both this last aspect and a detailed explanation of how each component has been modeled should be included. 

The conclusions are very brief and poor, as a reader I assume that the improvements have been presented in graphs but that does not prevent the conclusions from being more extensive and precise. 

Finally, I understand the difficulty of experimental verification even in cases like in my department where there is an anechoic chamber, but in spite of this, I find it difficult to accept papers that do not contain at least one prototype, however simple it may be. I would appreciate it if the authors could explain the limitations or assumptions involved in taking for granted the results of a simulation where not all physical effects can be modeled. 

Overall, I think the authors demonstrate a good knowledge of the subject matter presented, but the article needs to undergo a major revision. 

Round 2

Reviewer 2 Report

I have reviewed the new version of the article and as far as my concerns regards I must say that the authors have made all the modifications I indicated, improved the introduction and extended the explanation of the subject under study, further detailed the simulation, and most importantly compared their study with others, showing better results. I thank the authors for their effort and, for my part, no further improvement is necessary. I believe that the article can be published in its present form.